Comparison of soil bacterial community and functional characteristics following afforestation in the semi-arid areas

Deng Jiaojiao 1 2
Zhang Yan 1 2
Yin You 1 2
Zhu Xu 3
Zhu Wenxu zhuwx@syau.edu.cn 1 2
Zhou Yongbin 13998160246@163.com yyzyb@163.com 1 2
1 College of Forestry, Shenyang Agriculture University , Shenyang , China
2 Research Station of Liaohe-River Plain Forest Ecosystem, Chinese Forest Ecosystem Research Network (CFERN), Shenyang Agricultural University , Tieling , China
3 LliaoNing JianZhu Vocational College , Liaoyang , China
Marino Bruno
Electronic publication date: 2019 Jun 25
Publication date: 2019
Volume: 7
Electronic Location ID: e7141
Received 2019 Jan 21; Accepted 2019 May 17
Copyright: ©2019 Deng et al.
Copyright year: 2019
Copyright holder: Deng et al.
License: This is an open access article distributed under the terms of the Creative Commons Attribution License, which permits unrestricted use, distribution, reproduction and adaptation in any medium and for any purpose provided that it is properly attributed. For attribution, the original author(s), title, publication source (PeerJ) and either DOI or URL of the article must be cited.
License URL: https://creativecommons.org/licenses/by/4.0/

Keywords: Soil bacterial community, Bacterial functional characteristics, Afforestation, Thesemi-arid areas

Funding: Forestry Scientific Research in the Public Interest No.201404303-05 National Science and Technology Support Program of China 2015BAD07B30103 Sub-project of the National Key Research and Develepment Program 2017YFC050410501 Special Fund for Forest Scientific Research in the Public Welfare 201304216 Cfern & Beijing Techno Solutions Award Funds on Excellent Academic Achievements This research was financially supported by the special fund for Forestry Scientific Research in the Public Interest (No.201404303-05), the National Science and Technology Support Program of China (2015BAD07B30103, the Sub-project of the National Key Research and Develepment Program (2017YFC050410501), the Special Fund for Forest Scientific Research in the Public Welfare (201304216), and Cfern & Beijing Techno Solutions Award Funds on Excellent Academic Achievements. The funders had no role in study design, data collection and analysis, decision to publish, or preparation of the manuscript.

==============================
Changes in soil bacterial communities, which are crucial for the assessment of ecological restoration in Chinese plantations, have never been studied in the “Three North Shelterbelt” project in the semi-arid areas. We used high-throughput sequencing of the 16S rDNA gene to investigate the soil bacterial community diversity, structure, and functional characteristics in three plantation forests, including Populus × canadensis Moench (PC), Pinus sylvestris var. mongolica (PS), and Pinus tabuliformis (PT). In addition, soil environment factors were measured. There were distinct differences in soil characteristics among different plantation forests. Compared to PS and PT, PC had a higher soil pH, dissolved organic carbon (DOC), and available P, as well as a lower C/N ratio. Furthermore, afforestation with different tree species significantly altered the abundance of Proteobacteria, and Chloroflexi in the soil, and its influence on the bacterial diversity indices. The bacterial community compositions and functional groups related to C and N cycling from PS, and PT were grouped tightly, indicating that the soil bacterial phylogenetic distance of PS and PT were closer than that between PS plus PT and PC. Our results implied that the soil characteristics, as well as the diversity, compositions and functions related to C and N cycling of soil bacterial community obviously differed from the following afforestation, especially between PC and PS plus PT, which in turn enormously established the correlation between the soil microbial community characteristics and the afforestation tree species.

Introduction

Desertification has always been an important global ecological environmental problem in the arid and semi-arid regions (Li et al., 2004; Torres et al., 2015; Becerril-Piña et al., 2015), that is mainly caused by climate change and human activities in arid, semi-arid and some sub-humid regions (Chasek et al., 2015; Salvati et al., 2015; Wijitkosum, 2016), impacting 25% of the total terrestrial area (Reynolds et al., 2007; Allington & Valone, 2010). Desertification has caused a loss of soil nutrients, a decline in land productivity and environmental degradation (Li et al., 2018). This leads to a decline or degradation in sand-stabilization, soil conservation, water resource regulation, carbon sequestration and other desert ecosystem services, and it endangers both regional and national economic, social, and environmental security (Martã-Nez-Valderrama et al., 2016; Sutton et al., 2016). Research sponsored by the United Nations Environment Programme (UNEP) shows that the global economic losses caused by desertification and drought are as high as $4. 2 × 1010 USD per year, which is equivalent to all official aid given to Africa in 2009 (United Nations Convention to Combat Desertification (UNCCD), 2011) (Furtado & Macedo, 2008). With rapid economic growth, China is also confronted with various environmental problems, including sandstorms, severe desertification, and land degradation in dry northern regions (Liu & Diamond, 2005).

Numerous researches in arid and semi-arid areas have suggested that afforestation is one of the most commonly used techniques and effective sand-fixing methods to reduce the harm of desertification (Gao et al., 2002; Nunezmir et al., 2015). In addition, afforestation exhibits a significantly positive feedback to the regional environment (Verón & Paruelo, 2010; Zhang et al., 2014; Peng et al., 2014), successfully combating the desertification and dust storms (Fan et al., 2014; Tan & Li, 2015). In order to prevent ecosystem degradation, artificial shrubs and trees have been widely planted in many degraded areas of China (Zhao et al., 2017). Since the 1950s, large desertification control projects have been implemented (Zou et al., 2002; Piao et al., 2005; Qadir, Qureshi & Cheraghi, 2010). Since 1978, a series of ecological restoration programs have been initiated in China to alleviate these increasingly serious environment problems and to restore degraded land (Zhang et al., 2016), primarily through afforestation and reforestation (Zhang et al., 2014), of which the ‘Three North Shelterbelt Development Program (TNSDP)’ is the largest afforestation program in the world (Li et al., 2012). This special program in China gives us the advantage of studying the benefits of afforestation in China.

Currently, planted forests in China account for about 31.8% of the forested area, which is the most of any country in the world (State Forest Administration, 2010). By 2008, afforestation of 1,511,700 hm2 of land had been completed in the TNSDP of Liaoning. In this area, 212,200 hm2 of degraded forest was covered, accounting for 13.97% of the preserved area. Great achievement has been made in the afforestation of the Horqin Sandy Land in northern China (Zuo et al., 2012; Zhao et al., 2014; Ge et al., 2015), and the ecological environment has been considerably improved. Furthermore, afforestation has increased soil organic carbon sequestration in soil (Deng et al., 2006; Zhou et al., 2014). During vegetation recovery, soil characteristics have also been identified as the main factors driving plant growth, plant production, and dune ecosystem function (Zuo et al., 2008; Zuo et al., 2009; Qiu et al., 2018). A variety of researchers have focused on the influences of vegetation on the soil water content (Yang et al., 2018) and soil characteristics (Deng et al., 2017).

Compared with the physical and chemical properties of soil, soil microorganisms are more efficient and dynamic indicators of soil quality (Van der Heijen, Bardgett & Straalen, 2010; Bridge & Spooner, 2001). Soil microorganisms play a vital role in soil processes, which can profoundly impact the main biogeochemical cycles of C and N, as well as provide protection to larger organisms through the formation of biofilms (Chatterjee et al., 2008; Ritz et al., 2009; Burton et al., 2010). Previous research suggests that the soil microbial distribution is regulated by the different revegetation types (Deng et al., 2019). In addition, soil characteristics play important roles in shaping the soil bacterial community diversity and structure in the terrestrial ecosystems (Liu et al., 2018). Extensive studies have indicated that tree species and soil characteristics influence the microbial community structure and compositions (Ding et al., 2017). Within the fragile local ecological environment, there is also a close correlation between the soil biological properties and vegetation types. Thus, it is of great significance to study the characteristics of soil microorganisms in different plantation forests used for sand fixation forest (Wang et al., 2016). This information will allow for a better understanding of how soil improvement effects sand fixation forests and prevents land desertification. Previous findings have reported that grassland afforestation changes the chemical properties and composition of the soil, as well as the ecological functions of the soil bacterial communities. And these effects of afforestation on the microorganisms have been modulated by changes in soil chemical characteristics (Wu et al., 2019). Moreover, the forest soil habitat is an efficient means to restore local vegetation and studying this area can shed new light on the distribution of local soil eukaryotic microorganisms in semi-arid areas (Zhao et al., 2018).

Northwest Liaoning has an arid climate, frequent gales, water shortage and low vegetation cover. It is also the key governance area of the TNSDP. In recent decades, Populus × canadensis Moench (PC), Pinus sylvestris var. mongolica (PS), and Pinus tabuliformis (PT) have become the main afforestation tree species and widely used for afforestation in the “Three North Shelterbelt” due to their strong stress resistance, ability to accelerate ecological rehabilitation and improve ecological stability. Previously, numerous studies in shelterbelt have focused primarily on the aboveground ecosystem, and the soil physicochemical characteristics (Wang et al., 2014; Zhou et al., 2016). However, variations in the soil bacterial communities, which are crucial for the assessment of ecological restoration in plantations in this area, have never been studied. Therefore, we selected a typical artificial shelterbelt in the semi-arid area of northwestern Liaoning as the study area. The soil bacterial community diversity, compositions and functional characteristics were analyzed for different plantation forests composed of Populus × canadensis Moench, Pinus sylvestris var. mongolica, and Pinus tabuliformis. We propose the following hypotheses: (1) the three different plantation forests harbor different soil bacterial diversity and community structures; (2) the functional groups related to C and N cycling differ between PC and PS plus PT; and (3) consistent differences occur in the soil bacterial community and functional characteristics following afforestation. This research provides a reference for vegetation restoration and sustainable management of artificial forests in this area. Furthermore, this study has important theoretical and practical significance for the selection of tree species used for sand fixation in semi-arid regions. Our results provide a scientific basis for the recovery of degraded soils.

Material and Methods

Sites description

Research was conducted at Fujia forest farm, Changtu County, northwest of Liaoning province (123°32′E∼123°55′E∼42°53′N∼43°21′N). This area is located at the southern edge of Horqin Sandy Land and belongs to the Liaohe alluvial plain. The soil texture is characterized by Arenosols (Lv et al., 2018). The study area is located at an attitude of 91.1∼173.4 m, with the relatively flat topography and a small amount of elliptic or circular dune distribution. The climate in this region is classified as a temperate semi-humid semi-arid continental climate (Lu et al., 2017), with long and cold winters, and hot summers. There is little rainfall, with an average annual precipitation of 400 to 550 mm that is concentrated in July to August. The average annual evaporation is approximately 1,843 mm. The average temperature is 7 °C with extreme maximum and minimum temperature of 35.6 °C and −31.5 °C. In recent decades, Populus × canadensis Moench, Pinus sylvestris var. mongolica, and Pinus tabuliformis were the main plantation forests in the “Three North Shelterbelt” (Fig. 1). Prior to afforestation, the vegetation type was Arachis hypogaea (peanut) farmland, and the site information was shown in Table 1.

Soil sampling

Three independent replicate plots (20 m ×20 m) within the same climate were established in August 2018 for each forest type. The distance between each sampling plot was greater than 50 m but less than 200 m. To ensure the representativeness of soil samples, 10–15 soil cores of topsoil (0∼10 cm) were collected for each triplicate plot using soil auger with an “S” shape. After removing the litter layer, the soil cores were combined to one composite sample, giving a total of nine samples. All soil samples were stored on ice box after being sealed in plastic bags for transport to the laboratory. In the laboratory, the samples were sieved through a two mm mesh to remove plant roots, stones, litter, and other debris. Samples were subsequently divided into three parts. One part was air-dried for analysis of soil characteristics, including soil pH, the contents of total carbon (C), total nitrogen (N), and available phosphorus (P). The second part was stored at 4 °C for DOC analysis, and the third part was stored at −80 °C for DNA extraction.

Figure 1 Location of three forest type sites.

(A) Stusy site; (B) Populus × canadensis Moench; (C) Pinus sylvestris var. mongolica; (D) Pinus tabuliformis..

Measurement of soil characteristics

The soil pH was analyzed using an electrode pH meter in the soil-water (1:5 w/v) suspension (Bao, 2000; Ren et al., 2016). The contents of soil total C and total N were determined using an elemental analyzer (Elementar, Hesse, Germany) (Schrumpf et al., 2011). The concentrations of available P was determined using the extraction-flame photometry with a 0.5 M NaHCO3 extraction (Emteryd, 1989). Additionally, the content of dissolved organic carbon (DOC) was extracted from fresh soil using deionized water (1:5 w/v) (Gong et al., 2009) and determined via a TOC analyzer (Multi N/C 3100, Analytik Jena AG).

Table 1 Site information.

Different samples	Age of stand	Stand density (plant hm−1)	Height (m)	Diameter at breast height (cm)	Crown density (%)	
PC	18	773	15.34	14.32	65%	
PS	33	642	14.52	22.43	70%	
PT	33	575	13.56	20.51	65%	
Notes.

PC Populus × canadensis Moench

PS Pinus sylvestris var. mongolica

PT Pinus tabuliformis

Soil DNA Extraction and 16S rDNA Sequencing

DNA was extracted from 0.5 g of soil using the FastDNA SPIN Kit for Soil (MP Biomedicals, Santa Ana, CA, USA), following the manufacturer’s instructions. A NanoDrop ND-1000 spectrophotometer (Thermo Fisher Scientific, Waltham, MA, USA) was used to determine the quantity and quality of the extracted DNA. PCR was performed to target V3-V4 hypervariable region of the bacterial 16S rRNA genes was constructed to amplify, with the forward primer 338F (5′-ACTCCTACGGGAGGCAGCA-3′) and the reverse primer 806R (5′-GGACTACHVGGGTWTCTAAT-3′) (Deng et al., 2018). PCR amplifications were carried out in two steps. Firstly, each of three independent 25 µl reactions per DNA sample contained five µl of Q5 High-Fidelity GC buffer (5 ×); five µl of Q5 reaction buffer (5 ×); one µl (10 uM) of forward primer, one µl (10 uM) of reverse primer; 0.25 µl (5 U/µl) of Q5 High-Fidelity DNA Polymerase; two µl of dNTPs (2.5 mM); two µl of DNA Template (40-50 ng); and 8.75 µl of ddH2O (Deng et al., 2018). Cycling conditions were as follows: one cycle of denaturation at 98 °C for 5 min; then denaturation at 98 °C for 15 s, annealing at 55 °C for 30 s, and extension at 72 °C for 30 s (25 cycles); and with a final extension at 72 °C for 5 min. Agencourt AMPure Beads (Beckman Coulter, Indianapolis, IN, USA) and PicoGreen dsDNA Assay Kit (Invitrogen, Carlsbad, CA, USA) were used to purify and quantify PCR amplicons. Amplicons were then pooled at equal concentrations after the individual quantification step, and sequencing of pair-end 2 × 300 bp was performed using the Illlumina MiSeq platform with the MiSeq Reagent Kit v3 (Shanghai Personal Biotechnology Co., Ltd, Shanghai, China).

Functional prediction using FAPROTAX

The database of Functional Annotation of Prokaryotic Taxa (FAPROTAX), which is based on the available functional information in the existing microbiology literature, summarize the names of related species from functional classification and annotation information. FAPROTAX can be used to extrapolate functions of cultured prokaryotes to estimate metabolic or other ecological relevant functions, which is more suitable for the functional annotation and prediction of the biogeochemical cycle of environmental samples. The annotated operational taxonomic unit (OTU) table from the Silva database was read, and the annotated OTU information was matched with the species information in the database using a python program and the predicted functions were outputted. The details of this approach are provided by Louca, Parfrey & Doebeli (2016), Louca et al. (2017). The relative abundances of the functional groups in each sample was calculated as the cumulative abundance of OTUs assigned to each functional group, which was obtained by normalizing the cumulative abundance of OTUs correlated with at least one function. Thus, functional annotation of the OTUs was established based on FAPROTAX. We then investigated potential functions involved in geographical location and environmental conditions.

Bioinformatics and processing of sequencing data

The QIIME software (v1.9.0) and the UPARSE pipeline (Zhong, Yan & Shang, 2015) were used to analyze the raw data obtained from Illumina sequencing. The bacterial raw data was submitted to the NCBI Sequence Read Archive (SRA) under accession number PRJNA495735. The operational taxonomic assignment of OTUs with similarities >97% was conducted using the UPARSE pipeline (Edgar, 2013). Then the operational taxonomic classification and identity of OTUs were determined using a BLAST algorithm against sequences within the Silva Database via QIIME software (Kõljalg et al., 2013). OTU-level alpha diversity indices, such as Simpson index, Chao1 index, Shannon index, and ACE index were computed using the OTU table in QIIME (Caporaso et al., 2010).

Statistical analysis

Among samples, the unique and shared OTUs of the soil bacterial community were used to create Venn diagrams using the R (R v.3.4.4) with the “VennDiagram” package (Zaura et al., 2009). The heatmap representation of the top 50 classified bacterial genera in each sample was established using R (R v.3.4.4) with the “gplot” and “pheatmap” packages (R Development Core Team, 2009). The relationships between soil characteristics and the bacterial community functions related to C and N cycling based on Spearman’s rank correlation analysis were built using R (v.3.4.4) with the “psych” and “corrplot” packages (R Development Core Team, 2009). The differences in bacterial community structure across samples were established through beta diversity analysis and visualized via nonmetric multidimensional scaling (NMDS) based on unweighted UniFrac distance metrics (Lozupone et al., 2007).

Multifactorial ANOVA (MANOVA) and Canonical correspondence analysis (CCA) were applied to analyze the effects of tree species on all the measured soil characteristics. Multiple comparisons of means at a 95% confidence interval were performed using Tukey’s honest significance difference (HSD) post-hoc test. Soil bacterial diversity and relative abundances were analyzed in SPSS (v. 19.0) using a one-way analysis of variation (ANOVA) and least significant difference (LSD) multiple comparison tests (Banerjee, 2016). Spearman’s rank correlation was used to estimate the relationships between soil characteristics and soil bacterial community diversity. Similar patterns of functional groups of bacterial community were analyzed with Principal component analysis via the STAMP software (Parks et al., 2014). The linkages between soil environmental factors and bacterial community composition at the phylum level were performed by canonical correspondence analysis (CCA) via Canoco 4.5 (Braak & Smilauer, 2002).

Results

Soil characteristics in different plantation forests

Tree species had a strong significant effect on all the soil pH (F = 6.58, P = 0.031), total C (F = 30.54, P = 0.001), DOC (F = 6.02, P = 0.037), total N (F = 6.47, P = 0.032), C/N (F = 45.49, P < 0.001), and available P (F = 78.06, P < 0.001) (Table 2). There were distinct differences in the soil chemical characteristics among the three plantation forests. The soil pH ranged from 5.53 to 5.92. The highest pH value occurred in PC with 5.92, and significantly higher than PS and PT (P < 0.05). The soil DOC was highest in PC with 105.46 mg kg−1, and significantly higher than in PS and PT (P < 0.05). Soil total C and total N concentrations in PS were the highest with 12.00 g kg−1, and 1.05 g kg−1, respectively, followed by PT. And both of them occurred significantly higher than PC (P < 0.05). The maximum value of the C/N ratio occurred in PT with 11.79, followed by PS with 11.41, and lowest in PC with 9.15. No significant differences were observed in the soil total C content and C/N ratio between PS and PT (P > 0.05). Soil available P was highest in PC with 16.00 mg kg−1, which was significantly higher than those of PS and PT (P < 0.05) (Table 2). Compared to PS and PT, the PC had the highest soil pH value, DOC, and available P, as well as the lowest C/N ratio.

Table 2 Results of MANOVA and post-hoc analyses on the effects of tree species and their interactions on soil properties.

Factors	pH	Total C (g kg−1)	DOC (mg kg−1)	Total N (g kg−1)	C/N ratio	Available P (mg kg−1)	
Tree species	–	–	–	–	–	–	
F-value	6.58	30.54	6.02	6.47	45.49	78.06	
P-value	0.031	0.001	0.037	0.032	<0.001	<0.001	
PC	5.92a	7.82b	105.46a	0.85b	9.15b	16.00a	
PS	5.57b	12.00a	83.42ab	1.05a	11.41a	3.77b	
PT	5.53b	11.46a	80.10b	0.97a	11.79a	3.09b	
Notes.

PC Populus × canadensis Moench

PS Pinus sylvestris var. mongolica

PT Pinus tabuliformis

Different small letters meant significant difference at 0.05 level.

The first two axes of the CCA accounted for 84.3% of the total variance. The CCA plot showed a clear separation in the space among the three plantation forests. In fact, the PC distinctly separated from PS, and PT, especially along CCA1 (Fig. 2). The investigated soil characteristics also clearly separated into the quadrants. Soil total N was situated in the first quadrant; available P and soil pH were in the second quadrant; soil pH was in the third quadrant; soil total C and C/N ratio were in the fourth one (Fig. 2). The results illustrated that the different forest types had different soil characteristics, significantly different between PC and PT plus PS.

Figure 2 Results of canonical correspondence analysis-plot of all the measured soil environment factors.

PC: Populus × canadensis Moench; PS: Pinus sylvestris var. mongolica; PT: Pinus tabuliformis. TC: total C; DOC: Dissolved organic carbon; TN: total N; AP: available P.

Soil bacterial community diversity under different plantation forests

There were significant differences among different plantation forests regarding soil total C and total N contents ACE index and Chao1 index (F = 7.64, P = 0.02; F = 7.92, P = 0.02; Table 3). The maximum values of the ACE index and Chao1 index occurred in PS with 4,074.80, and 3,952.39, respectively, followed by PT and PC. The soil bacterial Shannon index and Simpson index among three plantation forests significantly differed (F = 6.89, P = 0.03; F = 5.68, P = 0.04; Table 3). The Shannon index and Simpson index were lowest in PT with 10.09 and 0.996, respectively. The Spearman’s rank correlations indicated that soil total N significantly positively correlated with the Chao 1 index (r = 0.78, P < 0.05) and ACE index (r = 0.83, P < 0.01). The Simpson index was significantly positively related with soil available P (r = 0.77, P < 0.05), and DOC contents (r = 0.67, P < 0.05), while, the Simpson index was highly negatively correlated with the C/N ratio (r =  − 0.73, P < 0.05) (Table 4).

Table 3 Soil bacterial diversity indices in different plantation forests.

Diversity indices	PC	PS	PT	F test	P value	
No. of sequences	48,848 ± 3,650aA	51,532 ± 2,892aA	52,395 ± 2,529aA	1.10	0.39	
OTUs number(Phylum)	2,750 ± 349bA	3,378 ± 150aA	2,837 ± 141bA	6.36	0.03	
Shannon index	10.46 ± 0.11aA	10.48 ± 0.16aA	10.09 ± 0.16bA	6.89	0.03	
ACE index	2,898.70 ± 537.94bB	4,074.80 ± 181.79aA	3,192.86 ± 344.77bAB	7.64	0.02	
Chao1 index	2,876.13 ± 518.02bB	3,952.39 ± 67.89aA	3,140.32 ± 290.58bAB	7.92	0.02	
Simpson index	0.998 ± 0.000aA	0.997 ± 0.001abA	0.996 ± 0.001bA	5.68	0.04	
Notes.

Data are means ± standard error (n = 3).

PC Populus × canadensis Moench

PS Pinus sylvestris var. mongolica

PT Pinus tabuliformis

Different small letters meant significant difference at 0.05 level. Different capital letters meant significant difference at 0.01 level.

Table 4 Spearman’s rank correlations between the soil bacterial diversity indices and available edaphic factors.

	pH	Total C	DOC	Total N	C/N ratio	Available P	
Simpson index	0.65	−0.65	0.67*	−0.55	−0.73*	0.77*	
Chao1 index	−0.48	0.55	−0.35	0.78*	0.25	−0.43	
ACE index	−0.58	0.62	−0.43	0.83**	0.30	−0.45	
Shannon index	0.22	−0.18	0.27	−0.09	−0.35	0.20	
Notes.

* correlation significant at 0.05 level.

** correlation significant at 0.01 level (two-tailed).

Soil bacterial community structure in different plantation forests

After quality trimming and chimera removal, 48,848, 51,532, and 52,395 high-quality sequences were generated from the PC, PS, and PT sites, respectively. Rarefaction curves for all the soil samples were shown in Fig. S1. As shown in the Venn diagram (Fig. 3), the total number of shared bacterial OTUs in PC, PS, and PT was 1,355. The number of bacterial OTUs shared between two sites was 1,503 for PT and PS, 744 for PS and PC, and 253 for PT and PC. The unique OTUs harbored in PC, PS and PT were 1486, 764 and 990, respectively.

Figure 3 Venn diagram representation of shared and unique OTUs of soil bacteria across three different plantation forests.

PC: Populus × canadensis Moench; PS: Pinus sylvestris var. mongolica; PT: Pinus tabuliformis.

Sequences analysis showed a total of 29 phyla, and 793 genera within the three plantation forest samples. Nine dominant phyla (relative abundance >1%) were observed, of which the total average relative abundances represented more than 95%. Proteobacteria was the most dominant bacterial phylum (31.01%), followed by Actinobacteria (23.76%), Acidobacteria (20.08%), Gemmatimonadetes (9.17%), Chloroflexi (8.07%), Firmicutes (1.52%), Planctomycetes (1.45%), Bacteroidetes (1.18%), and Verrucomicrobia (1.07%) (Fig. 4). The average relative abundances of the Proteobacteria subgroups (Alpha-, Beta-, Gamma-, and Delta-Proteobacteria) were 18.90%, 5.45%, 3.10%, and 3.56%, respectively (Fig. S2). The relative abundances of Proteobacteria and Chloroflexi varied significantly (P < 0.05) among the different forest types, with the highest abundances in PC with 34.78% and 10.02%, respectively (Fig. 4A). No significant differences were observed for other phyla among the different plantation forests (P > 0.05).

Figure 4 The relative abundance of dominant bacterial phyla (A) and genera (B) in different plantation forests.

PC: Populus × canadensis Moench; PS: Pinus sylvestris var. mongolica; PT: Pinus tabuliformis. Different small letters meant significant difference at 0.05 level.

At the genus level, 13 dominant bacterial genera (relative abundances >1%) were observed, namely RB41, Gemmatimonas, Sphingomonas, Crossiella, Jatrophihabitans, Variibacter, Rhizomicrobium, Pseudomonas, Bryobacter, Nitrobacter, Candidatus-Solibacter, Haliangium, and Pseudonocardia, accounting for more than 20% of the total relative abundances (Fig. 4B). The average relative abundances of Jatrophihabitans, Rhizomicrobium, Bryobacter, Candidatus-Solibacter, and Haliangium were significantly different among the three plantation forests (P < 0.05).

Effects of tree species on the compositions of the soil bacterial community

A cluster heatmap analysis was used to analyze the differences in the bacterial community compositions among the three plantation forests at the genus level (Fig. 5). The relative abundance and distribution of soil bacteria in different plantation forests changed significantly. Results showed that the soil samples were divided into two groups: one group contained the PT and PS, and the other group was PC. In order to show the bacterial community structures of PC, PS and PT, NMDS plot based on the unweighted uniFrac metric was calculated (Fig. 6). The NMDS plot also showed that the samples were classified into two large groups, one group corresponding to the communities from the PS, and PT, and the other group from PC. The samples from the PS and PT were grouped tightly, indicating that they shared a high similarity in their bacterial compositions. Furthermore, the PS and PT tended to be separate from the PC, especially along NMDS1, which contributed to 45.17% of the variance. Both analyses demonstrated that different plantation forests had different influences on the soil bacterial communities. Moreover, the phylogenetic relationship of the PT and PS was closer than those between the PT plus PS and PC.

Figure 5 Heatmap and hierarchical cluster analysis based on the relative abundances of the top 50 genera identified in the soil bacterial communities.

The samples are grouped according to the similarity of each other, and the clustering results are arranged horizontally according to the clustering results. In the figure, red represents the genus with higher abundance in the corresponding sample, and blue represents the genus with lower abundance. PC: Populus × canadensis Moench; PS: Pinus sylvestris var. mongolica; PT: Pinus tabuliformis.

Effects of soil environment factors on the soil bacterial community compositions

Results of the CCA showed that soil bacterial community structure had significant correlations between and soil characteristics (Fig. 7). At the phylum level, the first ordination CCA axis (CCA1) was strongly correlated with pH (r = 0.71), total C (r =  − 0.87), DOC (r = 0.71), total N (r =  − 0.71), C/N (r =  − 0.85), and available P (r = 0.91), explaining 54.5% of the total variability of the bacterial community structures. Both axes together explained 81.5% of the variation (Fig. 7A). At the genus level, the first ordination CCA axis (CCA1) was strongly correlated with total C (r = 0.58), C/N (r = 0.76), and available P (r =  − 0.74), explaining 55.7% of the total variability of the bacterial community structures. Both axes together explained 76.5% of the variation (Fig. 7B). Thus, soil DOC, C/N, and available P were important variables that played vital roles in the shaping of the bacterial communities.

Figure 6 Non-metric Multidimensional scaling analysis (NMDS) based on unweighted Unifrac metric illustrating the soil bacterial community structure among different plantation forests.

PC: Populus × canadensis Moench; PS: Pinus sylvestris var. mongolica; PT: Pinus tabuliformis.

Figure 7 CCA of abundant bacterial communities at the phylum (A) and genus (B) level and soil chemical characteristics for soil samples from different plantation forests.

TC: total C; DOC: dissolved organic carbon; TN: total N; AP: available P.

The relative abundances of Proteobacteria (r = 0.83, P < 0.01) and Bacteroidetes (r = 0.78, P < 0.05) had significantly positive correlations with the DOC content. The relative abundances of Proteobacteria (r =  − 0.67, P < 0.05), Chloroflexi (r =  − 0.68, P < 0.05), and Bacteroidetes (r =  − 0.77, P < 0.05) were significantly negatively correlated with C/N. In contrast, the relative abundances of Proteobacteria (r = 0.68, P < 0.05), Chloroflexi (r = 0.78, P < 0.05), and Bacteroidetes (r =  − 0.70, P < 0.05) were positively correlated with available P. The relative abundance of Verrucomicrobia was dramatically negatively correlated with soil pH (r =  − 0.83, P < 0.01), while, Verrucomicrobia was significantly positively correlated with total C (r = 0.90, P < 0.01), and total N (r = 0.83, P < 0.01). The relative abundance of Planctomycetes showed a significantly positive correlation with C/N (r = 0.70, P < 0.05), and negative correlation with available P (r = 0.88, P < 0.01) (Table 5).

Bacterial functional annotation and distribution in different plantation forests

According to the classification annotation results of the 16S rDNA sequences, a total of 51 functional groups were identified using FAPROTAX. These functional groups contained 5063 OTUs, and OTUs per functional group were listed in Table S1. We examined 13 ecological functional groups related to the C cycling, including chemoheterotrophy, aerobic chemoheterotrophy, phototrophy, photoautotrophy, photoheterotrophy, anoxygenic photoautotrophy S oxidizing, anoxygenic photoautotrophy, cellulolysis, oxygenic photoautotrophy, methylotrophy, methanol oxidation, hydrocarbon degradation, and methanotrophy (Table S2). The PCA plot showed that the functional groups related to C cycling in PT and PS were separate from those of PC, especially along PCA1 (Fig. 8A). Additionally, we examined 12 ecological functional groups connected to the N cycling, including nitrification, nitrate reduction, nitrogen respiration, nitrate respiration, aerobic nitrite oxidation, aerobic ammonia oxidation, nitrogen fixation, nitrite respiration, nitrate denitrification, nitrite denitrification, nitrous oxide denitrification, and denitrification (Table S3). The PCA plot showed that the functional groups related to N cycling in PT and PS were separated from those of PC, especially along PCA2 (Fig. 8B), indicating that the functional groups of PC differed from those of PS plus PT. We performed the Spearman’s rank correlation analysis to explore the relationships between the microbial functional groups and the six key environmental variables (Fig. 9). Soil pH value, total C, total N, C/N, and available P were the main factors influencing the functional groups related to C cycling (Fig. 9A). Whereas, total C was the main factor influencing the functional groups related to N cycling (Fig. 9B).

Table 5 Spearman’s rank correlations between the relative abundances of dominant bacterial groups and available edaphic factors.

	pH	Total C	DOC	Total N	C/N ratio	Available P	
Proteobacteria	0.60	−0.55	0.83**	−0.58	−0.67*	0.68*	
Actinobacteria	−0.25	0.15	−0.55	0.18	0.23	−0.15	
Acidobacteria	−0.17	0.27	−0.05	0.28	0.23	−0.35	
Gemmatimonadetes	0.35	−0.25	0.55	−0.17	−0.32	0.20	
Chloroflexi	0.52	−0.63	0.55	−0.52	−0.68*	0.78*	
Firmicutes	−0.33	0.40	−0.53	0.33	0.38	−0.43	
Planctomycetes	−0.35	0.60	−0.40	0.33	0.70*	−0.88**	
Bacteroidetes	0.40	−0.32	0.78*	−0.28	−0.77*	0.70*	
Verrucomicrobia	−0.83**	0.90**	−0.50	0.83**	0.53	−0.58	
Notes.

* correlation significant at 0.05 level.

** correlation significant at 0.01 level (two-tailed).

Figure 8 PCA plot of functional groups related to C cycling (A) and N cycling (B).

PC: Populus × canadensis Moench; PS: Pinus sylvestris var. mongolica; PT: Pinus tabuliformis.

Figure 9 The relationships between soil bacterial functional groups related to C (A) and N (B) cycles and soil environmental factors.

TC: total C; DOC: dissolved organic carbon; TN: total N; AP: available P.

Discussion

Soil chemical characteristics following afforestation with different tree species

Afforestation with different plantation forests had significant difference in soil conditions. PC had the highest pH value, when compared to PS and PT (Table 2), which was similar to the study demonstrating that the soil in pine stands had a lower pH than the oak and birch tree stands (Yoshimura et al., 2008). This difference might be the result of higher litter acidity in coniferous forests (Augusto et al., 2002). We observed that soil DOC in PC was higher than that in PS and PT. Our results were consistent with a previous study stating that soil organic matter and nitrogen were higher in broadleaf forests than those in coniferous forests (Jiang et al., 2012). Previous findings had established that both tree species and afforestation time dramatically influenced soil characteristics (Kim et al., 2018; Kang et al., 2018). In our study, the C/N values decreased in the order of PT >PS >PC, which was consistent with a previous finding that coniferous forests (pine) soil contained more carbon and had a higher C/N ratio than broadleaf forests (Yoshimura et al., 2008). The potential role of different forest types in variation of soil C/N ratio was also supported by previous finding (Mcgroddy, Daufresne & Hedin, 2004). In summary, the soil characteristics following afforestation with different tree species in the same area exhibited obvious differences, especially between PC and PS plus PT.

The bacterial community diversity response to different plantation forests

Similar to the soil characteristics, the Chao 1 index, and ACE index in PC were significantly lower than those in PS and PT (P < 0.01). Simultaneously, here we found that the Simpson index and Shannon index existed significant differences among different plantation forests (F = 6.89, P = 0.03; F = 5.68, P = 0.04; Table 3). This result might be due to the differences in the chemical compositions and decomposition rate of the litter (Kang et al., 2018). The Spearman’s rank correlations illustrated the Chao 1 index and ACE index existed significantly positively correlated with soil total N, which was similar to the research from northeast China that reported that the H’ value was positively correlated with the total N (Hui et al., 2014). The Simpson index was significantly positively related to soil available P, while, a previous study suggested that there was no significant correlation between diversity indices and P content (Wang et al., 2018). These results verified that there were significant differences in soil bacterial diversity among different forest lands.

The bacterial community compositions response to different plantation forests

The abundances of dominant bacterial phyla varied among the different plantation forests. In our study, Proteobacteria was the most dominant group, which was similar to the findings from Chinese pine plantations on the Loess Plateau (Dang et al., 2017), while, the research from Wulai forest reported Acidobacteria was the dominant member (Lin et al., 2014). Owing to differences in the lifestyles of Proteobacteria and Acidobacteria, they can be used as indicators of nutritional status (Hartman et al., 2008). In our study, the relative abundance of Proteobacteria in PC was significantly higher than that in PS and PT. In addition, the relative abundances of Proteobacteria had significantly positive correlations with DOC content. Our results concur with previous findings establishing that the availability of carbon was positively related with the abundance Proteobacteria (Fierer, Bradford & Jackson, 2007). In our study, Alphaproteobacteria was the dominant taxa at the class level. This is in agreement with the results obtained from a boreal peatland in Central Finland (Sun et al., 2014).

The phylum Acidobacteria is abundant in various soil environments (Zimmermann et al., 2005; Araujo et al., 2012; Meng et al., 2013). In our study, the abundance of Acidobacteria was relatively higher in the soil communities of PS and PT than that in PC, which was consistent with a previous investigation that the relative abundance of Acidobacteria was relatively higher in the coniferous forest than those of broadleaf forest (Christianl et al., 2008). Soil pH is generally considered as a key factor in shaping bacterial community structures (Preem et al., 2012). Research has established that the relative abundance of the Acidobacteria is dramatically associated with acidic soils (Jones et al., 2009), and more specifically, when the pH is lower than 5.5, Acidobacteria abundance increases (Lauber et al., 2009). However, our results found that the soil pH had no relation to the relative abundance of Acidobacteria, which might be due to the narrow range of pH from 5.53 to 5.92 (Table 2). PC stand had the highest abundance of Proteobacteria and lowest abundance of Acidobacteria. In consideration of the comparatively higher relative abundance of Proteobacteria observed in the copiotrophic soils and the relatively higher Acidobacteria abundance obtained in the oligotrophic soils (Fierer, Bradford & Jackson, 2007), we suggested that the PC plantation improved the soil nutrient conditions with lower C/N value. Gemmatimonadetes was the dominant bacterial community in our research, previous research illustrated that Gemmatimonadetes has been found in arid soils, such as grassland, prairie, and pasture soil, as well as pine soils (Debruyn et al., 2011). In our study, the relative abundances of Chloroflexi, and Bacteroidetes were positively correlated with available P, and available P might be one of the important factors influencing the bacterial community. Identically, previous findings have indicated that the phosphorus content has an effect on community structures (Fierer, Bradford & Jackson, 2007; Bergkemper et al., 2016).

The results of clear differentiation provided by the heatmap (Fig. 5) and NMDS (Fig. 6) plots illustrated that significant differences in the bacterial community compositions were observed among PS, PT and PC. The soil bacterial communities of the PT and PS sites were similar to each other, indicating that the hierarchical clustering distance between two coniferous forests was shorter than the distance between the coniferous and broadleaf forests. Our results were agreement with previous study which have established that the compositions of the soil bacterial community in hardwood forest differed from those in conifer forests (Lin et al., 2011; Ushio et al., 2008), which could release different quality and quantity of litter and root exudates (Sauheitl et al., 2010). In addition, the compositions of soil bacteria between PS and PT were also different. These results confirmed our hypothesis that the three different plantation forests harbored different soil bacterial community diversity and structure, suggesting that afforestation tree species had correlation with the soil bacterial community, which was consistent with previous findings (Ren et al., 2016; Gunina et al., 2017).

The bacterial functional groups response to different plantation forests

The C and N cycle in the terrestrial ecosystem and its regulatory mechanism are the hot topics in the science research of soil ecology and global change ecology (Maia et al., 2010). It is known that soil bacterial communities play an important role in biogeochemical cycles (Jenkins et al., 2017). In our study, we examined 13 functional groups related to C cycling. In contrast, a previous study from a temperate deciduous broadleaved forest and a tropical mountain rainforest detected eight ecological functional groups connected with the carbon cycle (Wei et al., 2018). Soil nitrogen fixation, nitrification, denitrification, ammonification, and other major nitrogen transformation processes are mainly mediated by soil bacteria (Yoon et al., 2015). And the soil nitrogen cycle, especially the biological nitrification and denitrification processes, can affect the production and emission of greenhouse gases, such as CO2, CH4, and N2O (Gregorich et al., 2005). To some extent, the denitrifying community of bacteria plays a vital role in the soil nitrogen cycle, and the relative abundances of specific OTUs are more valuable in predicting community function (Bent et al., 2016). In our study, the functional groups of denitrification were significantly higher in the PS than PT and PC (P < 0.05). For soil bacterial function, the functional groups related to C and N cycling in PT and PS were distinctly separate from those of PC, indicating that the functional groups of the broadleaf forest differed from those of the coniferous forests. Different plantation tree species could distinctly affect the community compositions of decomposers (Kubartová et al., 2007). Due to the existence of functional gene redundancy, these functional profiles are observed among bacterial communities (Fierer et al., 2012). Different plantation forests affect soil characteristics (Bhatia, 2008), thereby causing the change in the soil microbial diversity (Nair & Ngouajio, 2012), and functional diversity (Zhang et al., 2007). As a result, we believe our work has broad implications for reforestation in the semi-arid areas.

Conclusions

Our results revealed that soil characteristics after afforestation with different tree species under the same climatic conditions showed dramatic differences, especially between Populus × canadensis Moench and Pinus sylvestris var. mongolica and Pinus tabuliformis. Compared to Pinus sylvestris var. mongolica and Pinus tabuliformis, the plantation of Populus × canadensis Moench increased the soil pH value, DOC content, and soil available P content, while the C/N ratio decreased. Furthermore, the soil bacterial community compositions, diversity, and functions are different among plantation types, especially for Populus × canadensis Moench and Pinus sylvestris var. mongolica plus Pinus tabuliformis.

The bacterial diversity indices and the relative abundances of Proteobacteria, and Chloroflexi in the soil significantly differed among plantation types. The bacterial community compositions and functional groups related to C and N cycling from Pinus sylvestris var. mongolica, and Pinus tabuliformis were grouped tightly, indicating that the phylogenetic distance for microorganisms under different plantation types could be divided into two groups, including Pinus sylvestris var. mongolica, plus Pinus tabuliformis, and Populus × canadensis Moench. Our results highlighted that the soil bacterial community compositions and functions obviously differed following afforestation, especially between Populus × canadensis Moench and Pinus sylvestris var. mongolica and Pinus tabuliformis, which in turn enormously established the correlation between the soil microbial community characteristics and the afforestation tree species. Moreover, the bacterial community structure and functions related to C and N cycling showed consistent differences among different plantation forests following afforestation in the semi-arid areas.

Supplemental Information

Figure S1 The rarefaction curves of all the soil samples. The x-coordinate represents the total number of sequences randomly selected from each sample; the vertical axis represents the number of OTU observed at the corresponding depth

The length of the curve reflects the size of the sample sequencing. The longer the curve, the higher the depth of the sequencing, and the greater the possibility of observing higher diversity. The flatness of the curve reflects the impact of the sequencing depth on the diversity of the observed samples. The flatter the curve indicates that the sequencing result is sufficient to reflect the diversity of the current samples, and the further increase of the sequencing depth is unable to detect a large number of new OTU that have not been discovered. On the contrary, it indicates that the diversity is not close to saturation, and further increasing the sequencing depth will help to observe more new OTU. PC: Populus × canadensis Moench; PS: Pinus sylvestris var. mongolica; PT: Pinus tabuliformis.

Click here for additional data file.

Figure S2 Average relative abundances of Proteobacteria populations at the class level at each site

PC: Populus × canadensis Moench; PS: Pinus sylvestris var. mongolica; PT: Pinus tabuliformis.

Click here for additional data file.

Table S1 OTUs per functional group. Number of OTUs assigned to each functional group, compared to the total number of taxonomically annotated OTUs

Click here for additional data file.

Table S2 Functional groups related to the carbon cycle

Data are means ± standard error (n = 3). PC: Populus × canadensis Moench; PS: Pinus sylvestris var. mongolica; PT: Pinus tabuliformis. Different small letters meant significant difference at 0.05 level. Different big letters meant significant difference at 0.01 level.

Click here for additional data file.

Table S3 Functional groups related to the nitrogen cycle

Data are means ± standard error (n = 3). PC: Populus × canadensis Moench; PS: Pinus sylvestris var. mongolica; PT: Pinus tabuliformis. Different small letters meant significant difference at 0.05 level. Different big letters meant significant difference at 0.01 level.

Click here for additional data file.

Additional Information and Declarations

Competing Interests

Author Contributions

Data Availability

The authors declare there are no competing interests.

Jiaojiao Deng conceived and designed the experiments, performed the experiments, analyzed the data, prepared figures and/or tables.

Yan Zhang contributed reagents/materials/analysis tools.

You Yin performed the experiments, prepared figures and/or tables.

Xu Zhu conceived and designed the experiments.

Wenxu Zhu conceived and designed the experiments, analyzed the data, contributed reagents/materials/analysis tools, authored or reviewed drafts of the paper, approved the final draft.

Yongbin Zhou performed the experiments, analyzed the data, prepared figures and/or tables, authored or reviewed drafts of the paper.

The following information was supplied regarding data availability:

The bacterial raw data are available at the NCBI: PRJNA495735.

https://www.ncbi.nlm.nih.gov/bioproject/PRJNA495735.

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
