# Peer review of "Comparison of soil bacterial community and functional characteristics following afforestation in the semi-arid areas"

_PeerJ, doi:10.7717/peerj.7141_

## Round 0.1 · original submission · Major Revisions

The reviewers have made a number of suggestions for revision of the manuscript. The lack of a proper control group (e.g., pre-afforestation or non-afforestation soil data) creates an unsatisfactory experimental design for this component of the research. For example, reviewer 1 states:
“Authors want to compare the changes in soil properties and soil bacteria after three-different tree plantations. However, the lack of control group (e.g) makes it difficult to eliminate the influence of other factors, so it is difficult to confirm that the differences of soil properties and soil microbiomes are caused by different afforestation tree species. The author needs to make a more convincing argument in the article.”
A suggestion in this case is to decrease the reliance on this result and describe the field conditions that imposed the limitation. A number of related suggestions have been made regarding the statistical analysis that must be addressed including clarification of the stated hypotheses and actual number of samples analyzed.

Reviewers have made suggestions regarding revision of figures including clarification and combinations of figures.

A number of specific suggestions or questions regarding the use of English and the phrasing of sentences requires careful attention in your revision.

Please address each point made by the three reviewers and clearly indicate the revisions on a track-change word document. We look forward to your revised manuscript.

Reviewer 1 ·

Basic reporting

1. Although I, like the authors, am not a native English speaker, and I don't like being constantly criticized for the poor English writing, I have to ask the authors here to examine carefully the wording and, more importantly, the logic of writing. I would like to correct all of them, but there are too many ‘tiny flaws’, so I would like to ask the authors to check the whole manuscript carefully before the next submission.
2. Populus ×canadensis is nomencaltured by Moench. Therefore, Moench should not be italic.
3. The figure legend in Fig. 3 is too brief to understand what the author is presenting. What are those numbers? The “Veen” diagram should be revised as the “Venn” diagram.
4. Fig. 4 and 5 should be combined as one.
5. Fig. 6: Explain what the numbers of the color scale are.
6. Fig. 7: Please redraw the figure so that the numbers on the coordinates are at the bottom and left of the graph.
7. Fig. 9A and B should be typesetted as consistently as possible.
8. Figure 10 is too complex to read. Firstly, figure legend should be written more clearly and completely. Secondly, blue and red already indicate positive and negative correlation, so there is no need to use ellipse circle to express it. Third, for the part of the number, authors should present the “significance” (P-value) rather than the correlation coefficient (which has been expressed in color). And it would be easier to compare numbers if they could be placed on color squares rather than diagonally.

Experimental design

1. The sample sizes for soil property microbial 16S rRNA metagenome should be added. In L151, authors described that there are 9 plots with 10-15 soil cones per plots, so there should be at least 90 soil samples. However, Table 1 and 2 only showed three samples (n=3) in each plantation forest. Are those soil samples pooled together for physiochemical property analyses and microbial metagenome sequencing, or analyzing and sequencing independently but pooled data for analyses? For the former, many statistics are meaningless because of insufficient sample size; for the latter, all data should be re-analyzed separately.
2. A similar question as above, how many samples were used for the correlation analyses?
3. Authors want to compare the changes in soil properties and soil bacteria after three-different tree plantations. However, the lack of control group (e.g. pre-afforestation or non-afforestation soil data) makes it difficult to eliminate the influence of other factors, so it is difficult to confirm that the differences of soil properties and soil microbiomes are caused by different afforestation tree species. The author needs to make a more convincing argument in the article.
4. Some statistic suggestions:
(1) L211, L223, and Table 3 & 4: Using nonparametric correlation analysis, such as the Spearman, to replace the parametric Pearson correlation, unless you can prove that the data fit the assumption of a normal distribution (e.g. by Shapiro-Wilk normality test). In fact, I would encourage the authors to re-analyze all independent factors at the same time with GLM, rather than test the correlation of a single factor to a single response individually. If necessary, even consider multicollinearity and model selection. Using individual independent variables to test correlation without considering other factors is more suitable for the analysis of results under a controlled experimental environment rather than under natural conditions.
(2) L214, L221, L225: Authors used three kinds of ordination analyses to interpret relationships among planted tree species, edaphic factors, and microbial diversity. However, such analyses are messy and would be confusing. According to the research questions, I suggest using the constraint ordination analyses (CCA, RDA, etc.) to replace the unconstraint ordination analyses (NMDS, PCA, etc.).
(3) As for the testing of tree spp effect on edaphic characteristics, the PCA can just represent the result of “vector fitting” instead of testing the effect. When you say that you want to test the effect of X on Y, we would like to see can those dependent variables (Ys) be explained significantly by the independent factor (X) and how many proportions of variations can be explained. The vector fitting cannot tell you these stories. I suggest authors performing the MANOVA or PERMANOVA if you want to see how these tree spp. affect the entire soil environment, or you can use logistic regression to test the tree spp. effect on the influence of every single edaphic factor.
(4) Another analytical suggestion is that when you want to test the effect of an independent variable (such as a soil factor) on relative abundance (such as the RA of soil bacteria or functional groups), you need to consider the interference of richness. This is because RA is usually influenced by richness. Therefore, it is necessary to condition on "number of functional groups within soil samples" when testing the effect of edaphic factors on RAs of functional groups.

Validity of the findings

1. As I mentioned above, the insufficient sample size is a big problem. This leads to a lack of confidence in the outcome. For example, the conclusion “the influence of afforestation is greater on richness than on diversity (?) of soil microbes” (L369-371) is inconvincible because of the sample size problem. I can't offer any other suggestions here except to ask for an increase in the sample size unless authors can provide evidence to convince readers (and reviewers) that such sampling is representative.
2. The hypotheses are cliched despite clearly. I have some suggestions on these hypotheses:
(1) The first hypothesis is not a hypothesis. The comparison is a kind of methodology but a hypothesis. Unless the authors meant that these forests harbored different soil bacterial diversities and community structures. This hypothesis must be rewritten.
(2) The second and third hypotheses have already been common senses actually. I do not mean that these hypotheses cannot be tested here, but strongly suggest to provide more background information why these hypotheses should be tested in the Three North Shelterbelt afforestation areas. Is it expected to have different results from other studies, or is the change in microbial diversity and ecological function in the Three North Shelterbelt afforestation areas more important than other places and must be tested?
(3) I also suggest that some relevant literatures should be supplemented to strengthen the foundation of these hypotheses, for example, PeerJ 7:e6147 (2019), PeerJ 6:e6042 (2018), PeerJ 6:e5747 (2018), etc. In addition to the introduction of Three North Shelterbelt afforestation, these literatures on how the composition and ecological function of soil microorganisms have been altered should also serve as a basis for hypothesis-building.
(4) Finally, I suggest you cite your own paper PeerJ 6:e6251 (2019) as a basis to arise these hypotheses.
3. No result can validate the third hypothesis directly. The results only present 1) soil properties are different between broadleaf forest and pine forest, and 2) soil bacterial composition and the corresponding functions are different among different plantation forests. However, a conclusion was made that the soil bacterial diversity is influenced by the plantation tree species through altering the soil properties. This conclusion was just addressed in L446-450 actually, which is a discussion or deduction rather than a consequence from the investigation, analysis, or statistical evidence. I strongly recommend a more conservative conclusion.

Additional comments

I still have some minor comments in addition to those of the above:
1. How to define the top 50 genera of soil bacteria? Are the top 50 means those of total soil samples or of each soil sample?
2. L239-240: Please explain it.
3. L241: How many explanatory proportions in PC1 and PC2?
4. L246-248: The first-two axes plot of PCA can only show the separation of PC plantation from the other two, but cannot directly imply the “great impact” of forest type on soil properties. The interpretation is suggested to be more conservative.
5. L282: P>0.05
6. L340: I do not understand this sentence. Please explain it or rephrase it.
7. L369-370: Richness is also an index of alpha-diversity.

·

Basic reporting

no comment

Experimental design

no comment

Validity of the findings

no comment

Additional comments

In manuscript entitled “Consistent variations of soil bacterial community and function
characteristics following afforestation in the semi-arid areas”, the authors investigated the changes of the soil bacterial community diversity, structure, and function under different plantation forests, which is relevance for “Peer J”. This study is valuable and the results of this study are crucial for the assessment of ecological restoration in Chinese plantations. However, the manuscript has many problems should to be considered.
My main concerns about the article are as follows:
1. Line73-74, the statement of “ ‘Three North’ Shelterbelt Development Program” should be corrected as “ ‘Three North Shelterbelt Development Program’ ”.
2. L86, the ‘Three North’ Shelterbelt is not the first time you refer in the text, so please use the short term. Also note the same problem as follows.
3. L94-95, “Compared with the physical and chemical properties of soil, soil microorganisms are more sensitive and rapid to the change of soil quality” is there any published work that shows that this is true for the study region? I think this would be more convincing with an appropriate citation from the literature.
4. L99, the statement of “What is more” should be corrected as “In addition”.
5. A section in the introduction on the silvics of and soil characteristics preferred by Pinus sylvestris (PS), Populus ×canadensis Moench (PC), Pinus tabuliformis (PT) would have helped the reader not familiar with the species. The soil description of the sites was adequate.
6. L136-137, Please, use classes based on international taxonomy classifications.
7. L304-306, CCA like multiple regression show an increase in the variance explained just by adding more variables. Many studies with CCA with similar number of points and variables show very high % of explained variance, but usually these results are spurious. So, how this axis could be interpreted? is showing any kind of gradient?
8. The manuscript contains many grammatical errors that need to be corrected, so a native English speaker review and revise the manuscript would be useful.

Reviewer 3 ·

Basic reporting

Overall, this paper has a professional English writing style, some places still need further editing, for example: line 35, replace “under” with “in”; Line 38-39, should not use “increased” because this is not a manipulation study like nutrient addition, please use “had higher soil pH, DOC…and lower C/N ratio”; Line 39, maybe write it out for DOC, dissolve organic carbon; Line 43, when you decided use C or carbon? Above authors had C/N but here authors had carbon and nitrogen, please be consistent; Line 45-46, this sentence is not very informative, it can be applied to anywhere, maybe delete it for Abstract; Line 51, if authors want this paper to be searched broadly, maybe use “semi-arid area” instead of “the semi-arid areas of Northwest Liaoning”; Line 125, no need for that “,”; Line 136, inconsistent font for those coordinates, that maybe why there is a big space there; Line 155, typo, “r”; Line 158, extra space; Line 229, 250, 254, 292, can just say "three plantation forests"; authors can use "different" in the Discussions section; Line 238, again, better not use "increase or decrease", using "had higher or lower.."; Line 328-330, odd space, please choose left align; Line 351, "were higher than those in the PC"; Line 354-355, long sentence, break it down; Line 361, avoid using "anyway", this is only for oral, should not use it in scientific writing. Probably say, "in summary" or "overall"; Line 457, "especially for?";

The first paragraph of Introduction made this paper fall into a limited pool of audiences. I suggested to start with discussing desertification issues globally, more general. And put the description of Chinese program in 1950s and 1978s in the last paragraph or second last paragraph with sentences like “the special program in China gives us an advantage of studying the benefit of afforestation…”

Line 80, this sentence is not correct: “increased soil organic carbon sequestration in both above ground biomass and soil”, aboveground biomass is not about soil organic carbon

Line 90, this first sentence sounds like the authors will be talking about soil characteristics but then the next few sentences are all about soil microorganism. So the topic sentence of this paragraph is confusion. I suggested starting the paragraph with the second sentence

Line 95, is “soil quality” the same thing as “soil characteristics”? and in line 97 “soil properties” and in line 102 “soil parameters” are those the same thing? Authors may avoid using too many different words to describe a similar thing

Line 104, what is “vegetation”? vague, coverage? Vegetation type?

Line 121, “16S rRNA high-throughput gene sequencing technique” can be omitted here, this belongs to Method section

Line 126, it is hard to make this judgement because authors only have 1 replicate for broadleaf (Populus Canadensis Moench) even if using NMDS

Line 127, better to use “difference” instead of “change”.

Line 130, if authors want to discuss the point of selecting tree species, should give more details about what tree species were commonly planted in the Introduction, only those three species reported here?

Figure 1, change panel B to Pinus sylvestris, panel C to Pinus tabuliformis, etc to be consistent with the order of description in the text

Figure 4, please indicate what "a, b" indicates in the figure caption.

Figure 4 and Figure 5, what is "PA"? typo? should be "PC"?

Table 3 and Table 4, two digits would be enough

Experimental design

Authors need some references in the section of site description, particular about soil type and climate.

What are those same conditions? Climate? Vague

Line 150, no up limit? For the plot distance

The section of “Measurement of the soil properties” missing description of quality control (reference materials, recovery, etc).

The section of “statistical analysis” should be separated into two paragraph, maybe at line 219

Validity of the findings

Line 239, why this sentence is here? "This work is guided on “Observation Methodology for Long-term Forest Ecosystem Research” of National Standards of the People's Republic of China (GB/T 33027-2016)." seems not relevant to the results

Line 262-263, this first sentence is not necessary, belong to method; choosing another topic sentence

Line 266, "Venn diagrams were used to compare.." method sentence, not belong to results

Line 304-308, those three sentences are method to me. "a clear correlation" what does that mean? Please choose a better topic sentence and directly describe about results. instead of saying using what method to plot what and in what figure.

Line 347, again, better not using "change", using "had difference" instead. Because technically, those three forest types were all manipulation plots, authors don't have a control with nothing planted to be compared with.

Line 356, what is the afforestation time in this study?

Line 454, in the conclusion, authors may want to write out the full name of PS, PT and PC to make it readable with itself.

Line 458, another paragraph from here?

Line 462, again, it is hard to make this conclusion because of only 1 replicate for broadleaf.

Additional comments

This paper describes variations in soil physical parameters, and soil bacterial community and function at three plantation forests in the semi-arid areas in China. This paper has important values for the studies of afforestation as it reports the basic soil characteristics. Please revise the paper based on the comments listed above.

---

## Round 0.2 · Major Revisions

While many revisions have been made as suggested by the previous round of reviews, concerns regarding the statistical significance of the results remain. For example, Reviewer 1 states: "I don't think it's very meaningful to evaluate the results of this study until the suggestions of these statistical analyses have been substantially answered and revised." Rather than reiterate the points raised here, I suggest that you address, point by point, the questions raised regarding the statistics.

Reviewer 1 ·

Basic reporting

This article has been revised according to some of the suggestions provided by three reviewers. However, I still have serious concerns about the statistical analyses which are the basis to make conclusions of the causes or effects on the soil bacterial diversity. Authors have hardly answered any questions about statistics positively. So some similar questions and comments are given this time. In addition, there are still multiple typos. Authors should carefully check the article before submission.

L132: “However, variations in the soil bacterial communities, which are crucial for the assessment of ecological restoration in Chinese plantations, have never been studied.” This sentence is incorrect. I can easily search for lots of papers about the community diversity of soil bacteria in China, and several of them are related to the afforestation. For example, Zhao et al. (2018) Eur J Soil Sci, 69: 370-379; Liu et al. (2018) Plant Soil 423: 327; Yang et al. (2018) European Journal of Soil Biology 85: 73-78; Wu et al. (2019) PeerJ 7: e6147.

Experimental design

A question raised in the experimental design in last comments, authors responded that they emphasize the differential bacterial community between forest types and focus on the soil characteristics and high-throughput sequencing (to estimate the bacterial community compositions). They also declared that the use of LEfSe analysis to determine the “classified bacterial taxa with significant abundance differences”. However, this response did not answer the question of no control group and how to avoid other confounding effects in addition to the forest types.

If authors want to identify the bacterial taxa with significant abundance differences between forest types, why not just simply comparing the abundances of sampled bacterial taxa with more direct and intuitive statistical methods? why choosing the LEfSe?

Validity of the findings

In statistics, the authors seemed not to understand what I meant. For example, in ordination analysis, I emphasize that unconstrained ordination analysis can only explain "vector fit" rather than "effect", in other words, it cannot tell any causal relationship by any unconstrained ordination analysis. So, if authors want to test some “effect”, please use the ordination analysis.

As for the MANOVA and/PERMANOVA, the tested effect is categorical data (e.g. tree species, which is the independent factor) and the response is edaphic characteristics (this is the dependent factor, i.e. the outcome or response), and the dependent factor is comprised of “multiple variables”, so the “multivariate” analysis of variance (MANOVA) or the permutational multivariate analysis of variance (PERMANOVA) is suggested. Strictly speaking, your experiment has not done any treatment. If you have other treatments (such as varying degrees of watering) beyond this research question (effect of planting species on edaphic characters), you need to set the watering factor as the random effect. If you are not familiar with relevant statistics, I strongly recommend consulting experts in statistics.

Another statistical question concerns the conditional factor. I asked the authors to condition on the richness when testing the effect of edaphic factors on RAs of functional groups of soil bacteria, instead of asking how to estimate the richness. The reason I ask this question is simple. If the total numbers of bacteria are the same in two communities, but there are only 10 bacteria in community A and 10,000 bacteria in community B, the relative abundance of the same bacteria in two communities will inevitably different by the differential bacterial richness. Therefore, the bacterial richness in each soil sample must be taken into account in estimating the effect of edaphic effects on microbial RA. Ignoring it may directly affect the results of estimates.

Additional comments

I don't think it's very meaningful to evaluate the results of this study until the suggestions of these statistical analyses have been substantially answered and revised.

·

Basic reporting

no comment

Experimental design

no comment

Validity of the findings

no comment

Additional comments

The authors have intensively modified the original manuscript following reviewers' comments and suggestions. They have also answered, explained and clarified the reviewers' questions and concerns. The improvement is significant and mostly satisfactory.

Reviewer 3 ·

Basic reporting

Line 78, The first sentence is not relevant.Omit it

Line 94, "...of the forested areas"

Line 168, "soil corns"? should it be "soil cores"?

Line 173, this is a laboratory analysis, so C/N should not be here. It is calculated by mathematics.

Line 278-279, Authors declared "no significant differences.." but with a p value less than 0.01? And for p values throughout the results section, I prefer to see the specific p values if there are. For example, if one p value is 0.02, authors should write "0.02" instead of "< 0.05". If one p value is 0.23, authors should write "0.23" instead of "> 0.05".

Line 313, the first sentence is method, no need to be here

Line 320, ",and the other..."

Line 332 "In total..." This sentence is not necessary, omit it

Line 334-335, would two digits for r values be enough?

Line 394, I am confused by this sentence: "The Simpson index of PC was the highest, while, there were no significant differences among different plantation forests (P < 0.01)" why saying no significant differences but the p value is less than 0.01?

Line 415, "previous findings that have established that a high availability of carbon has been associated with a high.." this sentence is bad, revise it

Line 424, "..considered as a key..."

Line 497-499, no funding source for this study?

Experimental design

None

Validity of the findings

None

Additional comments

Authors have revised this paper intensively based on reviewers' comments. I really appreciated that effort! I only have some minor suggestions on some sentences (see above).

---

## Round 0.3 · Minor Revisions

Reviewer #1 has emphasized the potential weakness in your use of statistical methods and the conclusions drawn from them. Rather than pursue details of the analysis, the reviewer suggests that you limit discussions of causality. I am in agreement with the conservative interpretation approach. Please pay careful attention to all the suggestions made in your revision.

The sentiment of Reviewer 1 is added here:

I strongly recommend referring to the article “Paulson et al. (2013) Nature Methods 10(12):1200–1202” for the identification of taxa with key differences between communities. As for other statistical suggestions, if the authors are not so familiar with the principles and applications of these analyses (unconstrained and constrained ordination analysis, MANOVA, linear regression, linear mixed-effects model, etc.), I suggest that the description and discussion of their results should be conservative, especially avoiding the discussion of causality.

We look forward to your revision.

Reviewer 1 ·

Basic reporting

NA

Experimental design

NA

Validity of the findings

NA

Additional comments

In fact, what I am most concerned about is the correctness of statistical methods and the interpretation of statistical results, because the inferences of this marker-based community ecological process almost depend on statistical results. I do not intend to argue with the authors about the statistical method. The importance of discriminating key microbial taxa by LEfSe has been repeatedly emphasized in rebuttal letter. I am sorry that I am not familiar with this method. However, as far as I understand, LEfSe is an application of linear discriminant analysis (LDA), while LDA can be regarded as a combination of ANOVA and regression analysis, and the fundamental assumption of LDA is the normal distribution of independent factors. If the independent variables cannot be assumed to be normally distributed, the use of ANOVA and/or regression analysis is more appropriate than using LDA (or LEfSe). I strongly recommend referring to the article “Paulson et al. (2013) Nature Methods 10(12):1200–1202” for the identification of taxa with key differences between communities. As for other statistical suggestions, if the authors are not so familiar with the principles and applications of these analyses (unconstrained and constrained ordination analysis, MANOVA, linear regression, linear mixed-effects model, etc.), I suggest that the description and discussion of their results should be conservative, especially avoiding the discussion of causality. In principle, I affirm the authors’ efforts and patience in data processing and research results in this study, although I think this data should deserve better interpretation. So I don't have any further critical comments, except for a few minor ones.

1. L382-385: This inference is too arbitrary and suggests weakening or deleting it.
2. L397: There are differences in soil composition and microbial diversity among different forest lands. However, due to the failure to detect before and after afforestation, it is difficult to verify whether the difference comes from the difference in geographical distribution or is really caused by afforestation. It is, therefore, necessary to avoid such causal inferences.
3. L428-432: As mentioned above, such an inference is too arbitrary. It is not difficult to find a positive or negative correlation between some specific microbial taxa and the content of some soil components in a large number of microbial community dataset, but there is not necessarily a causal relationship between them. It is suggested that inferences should be more conservative when false positives cannot be ruled out or the causal relationship cannot be evident.
4. L430: “Available P” should be “available P”.
5. L441-446, L471-473, L483-485: These inferences should be more conservative and avoid the inference of causality.
6. L475-476: Change to “The soil bacterial community compositions, diversity, and functions are different among plantation types, …”. L478-479 should also be modified in this way.
7. For the same reason, I don't think the experimental design and analytical method of this study can answer the causal relationship. At most, it is the correlation between the soil microbial composition/diversity/function and the afforestation tree species. Therefore, I suggest that the title of this article should be modified appropriately.

Reviewer 3 ·

Basic reporting

None

Experimental design

None

Validity of the findings

None

Additional comments

I think this paper is well revised, and I don't have any further comments. I suggest to accept this paper as it is.

---

## Round 0.4 · Minor Revisions

Thank you for the careful revisions. While both reviewers have indicated acceptance, there are several minor points to address prior to formal acceptance. Reviewer 1 notes:

1. Please check the scientific name carefully, especially spaces before and after the “’x” between the generic name and species name of hybrid species Populus × canadensis.
2. Fig. 9: there are two redundant correlation coefficient values between TN and TN and between TN and C/N.
3. L346-349: Please check the space carefully. There seem to be two spaces between some words.

We look forward to the final revisions.

Reviewer 1 ·

Basic reporting

NA

Experimental design

NA

Validity of the findings

NA

Additional comments

Overall, I'm satisfied with this version. In addition to some minor details that still need to be corrected, I would recommend accepting this article.

1. Please check the scientific name carefully, especially spaces before and after the “’x” between the generic name and species name of hybrid species Populus × canadensis.
2. Fig. 9: there are two redundant correlation coefficient values between TN and TN and between TN and C/N.
3. L346-349: Please check the space carefully. There seem to be two spaces between some words.

---

## Round 0.5 · accepted · Accept

Thank you for your attention to the final revisions suggested by reviewers. In accordance with the reviewer's acceptance of the manuscript pending revisions, the article is now accepted for publication in PeerJ.

# [#PeerJ Saff Note: Although the Academic Editor is happy to accept your article as being scientifically sound, a final check of the manuscript shows that it still needs another round of English editing. Please can we ask you to edit the language one last time, as we do not perform language editing as part of our production process#]